# OpenReview forum: "RCOT: Detecting and Rectifying Factual Inconsistency in Reasoning by Reversing Chain-of-Thought"
_ICLR.cc/2024/Conference — ICLR 2024 Conference Withdrawn Submission_

### Official Review · Reviewer_vr2D · 2023-10-22

**Soundness:** 2 fair
**Presentation:** 2 fair
**Contribution:** 2 fair
**Rating:** 5
**Confidence:** 5

**Summary:**

This paper aims to improve LLM’s reasoning abilities and address the challenges of overlooking, question misinterpretation and condition hallucination in LLMs’ generated solutions. It proposes RCoT to detect and rectify such factual inconsistency through four steps, including reconstruction, decomposition, comparison, and revision. The experiments are conducted on randomly sampled sub-sets of seven arithmetic datasets.

**Strengths:**

- The motivation is clear and the analysis of challenges is reasonable.
- The performance of the proposed RCoT is demonstrated to be superior to the standard baselines.

**Weaknesses:**

- The experiments are only conducted on randomly sampled sub-sets of the test sets, which may raise concerns about the convincingness of the results. The experiment results do not allow for a direct apple-to-apple comparison with the reported results in other papers, such as those related to Self-consistency.
- The experiments on other reasoning tasks, such as commonsense reasoning and symbolic reasoning, are absent.
- The performance improvement is not significant compared to the self-consistency (84.5 v.s 83.5).  In addition, did the paper's testing of the Self-consistency algorithm use 30 paths? In Self-consistency, the typical number of paths used is (1, 5, 10, 20, 40). Why were 30 paths chosen? If the reason is to make comparable comparisons based on average tokens, it would be appropriate to report the performance and average tokens under different numbers of paths.
- There is a lack of in-depth analysis and evaluation beyond overall performance, such as the absence of assessment regarding improvements (Quantitative or user-study-based evaluations) in the three areas of overlooking, question misinterpretation, and condition hallucination. Table 5 only evaluates on 45 cases.
- The method is somewhat incremental. The decomposition, comparison, and revision components are not new in the context of CoT. While reconstruction is used in many fields, its application within CoT is new and appears to be the main technical contribution. However, the overall framework of RCoT is incremental and complex.
- Minor suggestions about the presentation:
    - On page 1, this paper introduces the condition of  "2 days away" in Figure 1 is mistakenly overlooked. However, there are no "2 days away" in Figure 1.
    - Can the three different examples in the Introduction be unified?
    - The font size in Table 1 is too small.
    - The order of citing figures and charts is mixed up. For example, Table 4 is cited before Tables 2 and 3, but it is located below them in the paper.

**Questions:**

See Weaknesses

---

> ### Author Response · Authors · 2023-11-19
> **Response to reviewer vr2D (1/2)**
>
> **W1: only conducted on randomly sampled sub-sets of the test sets**
>
> Thanks for pointing out this problem! According to our observation, in asdiv, svamp, Date, addsub, singleeq datasets, 256 samples can well cover various types of problems. For the more diverse datasets aqua and gsm8k, we include the entire aqua dataset, and for gsm8k we expand the number of samples to 550 (see Figure 1). Considering that 3 groups of 256 can already cover all datasets as various types of problems and reduce budget overhead, so we conduct experiments on sub-datasets.
>
> We would also want to address that apple-to-apple comparison to results in other papers is hard, or sometimes, impossible. For example, Self-Consistency does not use ChatGPT, Active-Prompting uses CodeX which is not supported anymore, and Self-Refine uses gpt-3.5-turbo which changes over time. Therefore, we believe implementing them ourselves and guaranteeing them in a fair environment is a better choice.
>
>
> **W2: The experiments on other reasoning tasks, such as commonsense reasoning and symbolic reasoning, are absent**
>
> This a very interesting question! Currently, we only focus on math reasoning tasks due to the limited time. In the future, we will explore RCoT on other reasoning tasks. Thank you for your suggestions.
>
> **W3: Report the performance and average tokens under different numbers of paths**
>
> We apologize for the confusion. And we report more different numbers of paths(10 and 20 for self-consistency). See the result:
>
> | Method                                                                      |  GSM8K |  AQuA  | AddSub |  Date  | SingleEq |  ASDiv |  SVAMP |   Avg acc  | Avg tokens |
> |--------------------------------------------------------------------------|:------:|:------:|:------:|:------:|:--------:|:------:|:------:|:----------:|:----------:|
> | SC (10 trials per problem)                                      | 80.46% | 68.11% | 86.32% | 78.90% |  92.57%  | 88.67% |  78.1% |   81.88%   |   1800.2   |
> | SC (20 trials per problem)                                      | 81.64% | 69.29% | 88.28% | 79.68% |  92.57%  | 89.84% |  79.6% |   82.99%   |   3784.1   |
> | SC (30 trials per problem)                                      | 81.64% | 70.86% | 88.67% | 80.07% | 92.96%   | 90.23% | 80.47% | 83.56%     | 5615.0     |
> | RCoT (1 trial per problem)                                      | 82.03% | 56.29% | 87.20% | 71.87% | 92.4%    | 86.33% | 79.69% | 79.40%     | 1831.0     |
> | RCoT (3 trials per problem)                                     | 83.20% | 72.83% | 89.84% | 78.91% | 93.75%   | 91.80% | 81.25% | **84.51%** | 5453.3     |
> | Self-Refine attempt 0 (i.e., Standard CoT)                      | 79.12% | 45.28% | 90.62% | 51.38% | 97.65%   | 83.59% | 75.29% | 74.70%     | 190.2      |
> | Self-Refine attempt 1 (1 trials per problem * 2 call per trial) | 80.72% | 49.21% | 91.41% | 52.75% | 98.04%   | 84.37% | 76.86% | 76.19%     | 3108.4     |
> | Self-Refine attempt 2 (2 trials per problem * 2 call per trial) | 80.72% | 49.21% | 91.41% | 52.75% | 98.04%   | 84.37% | 76.86% | 76.19%     | 3324.9     |
> | Self-Refine attempt 3 (3 trials per problem * 2 call per trial) | 80.72% | 49.21% | 91.41% | 52.75% | 98.04%   | 84.37% | 76.86% | 76.19%     | 3359.6     |
> | Self-Refine attempt 4 (4 trials per problem * 2 call per trial) | 80.72% | 49.21% | 91.41% | 52.75% | 98.04%   | 84.37% | 76.86% | 76.19%     | 3367.7     |
> | Self-Refine attempt 5 (5 trials per problem * 2 call per trial) | 80.72% | 49.21% | 91.41% | 52.75% | 98.04%   | 84.37% | 76.86% | 76.19%     | 3367.7     |
>
> Although RCoT cannot be compared with self-consistency with similar costs, we highlight that combining RCoT with other methods can achieve better performance. RCoT mainly focuses on checking and revising LLMs’ first answer, which is different from other traditional methods.  Moreover, combining RCoT with existing methods can improve upper-bound performance. For example, we can observe from Table 1 and Table 4 that combining Self-Consistency and Active-Prompting methods with RCoT can further improve average accuracy by 5.1% and 2.1% across seven datasets, respectively.

---

> > ### Author Response · Authors · 2023-11-19
> > **Response to reviewer vr2D (2/2)**
> >
> > **W4: There is a lack of in-depth analysis and evaluation beyond overall performance**
> >
> > It’s a helpful suggestion. We admit that due to human labor and time, we only manually conducted human evaluation on 100 GSM8K problems and computed the recall in Table 5. It’s worth noting that we put many concrete examples to prove that RCoT can effectively detect factual inconsistencies. Specifically, Figures 6-9 show examples of overlooking, and Figures 10-16 show examples of hallucination. From the quantitative analysis, we can find that RCoT is not perfect, and we encourage more effective methods to be explored. Since the limited time, we cannot conduct many user-based studies. In the future, we will conduct deeper quantitative analysis and evaluate more samples.
> >
> > In our paper, we highlight the necessity of fine-grained feedback. We show that human feedback with coarse-grained feedback (86.3% Acc on GSM8k) is worse than human feedback with fine-grained feedback (94.6% Acc on GSM8k). The reason is that fine-grained feedback points out the concrete error, whereas coarse-grained only points out whether the overall response is correct, making it hard for LLMs to know where to revise. We will add this discussion with concrete examples in our future revision.
> >
> >
> > **W5: The method is somewhat incremental**
> >
> > We understand your confusion. The CoT method usually has a lower accuracy for more complex problems, while RCoT, a fine-grained method, has a higher improvement of performance. We encourage the NLP community to explore how to improve the feedback generated by LLMs as humans. See more detail in our general response W2.
> >
> >
> > **W6: Minor suggestions**
> >
> > Thank you very much for finding these errors. We have corrected them and once again apologize for the confusion.

---

> > > ### Author Response · Authors · 2023-11-22
> > > **Official Comment by Authors**
> > >
> > > Dear reviewer vr2D,
> > >
> > > We appreciate your effort in reviewing our paper! Please let us know if we address your questions, and we are willing to have further discussions if needed:)

---

### Official Review · Reviewer_rMGU · 2023-10-29

**Soundness:** 2 fair
**Presentation:** 2 fair
**Contribution:** 2 fair
**Rating:** 3
**Confidence:** 4

**Summary:**

The paper proposed a novel approach RCoT to identify and rectify factual inconsistencies in the outputs of large language models (LLMs). The approach works by first prompting the LLM to reconstruct the question based on its answers, and then prompting the LLM to determine whether the reconstructed question is identical to the original question in terms of the conditions derived. If there are discrepancies between the conditions, the finer differences are used to rectify the LLM to provide a more accurate and consistent answer. Experiments show that the proposed RCoT outperform baselines in seven arithmetic datasets.

**Strengths:**

The general thoughts of the problem are interesting and novel -- proof by contradiction -- use LLM to prove things by contradiction. From my understanding, LLMs are used in the following different scenarios: (1) reconstructing the problem; (2) Listing the conditions of the original and reconstructed problems; (3) determining whether there are hallucinated and overlooked conditions; (4) Determining whether the reconstructed problem is identical to the original one. (5) rectifying the results based on the finer feedback summarized from (3). The above uses of the LLM are interesting and each worth individual study about the effects.

**Weaknesses:**

Although the general thoughts of the problem are interesting and novel, the paper itself has many obvious flaws.

First, the prompting method is overused. The paper does not establish causal connections between the different prompting stages. For example, to determine whether the reconstructed problem is identical to the original one (Figure 20, “question comparison”), the method does not consider the prompted results from “problem decomposition” and “condition comparison”. Additionally, to rectify the solution, the model does not take the results from “question comparison” into account.

Second, upon reviewing the examples  (Figure 20 "I apologize for my mistake...". ), I believe that the authors are exploiting the "dialog system" nature of the LLM interfaces. The dialog system introduces an extra conditioning on the chat history, which means that the actual prompts listed in the paper are all conditioned on previous prompts that have been used. As a result, I believe that the experiments are flawed. In contrast, the methods such as CoT (Wei et al., 2023), Active Prompt (Diao et al., 2023), and Self-Consistency (Wang et al., 2023) are stateless, meaning that they only involve a single interaction between a human and the LLM interface. I recommend that the authors learn from CoT which models P(answer|question, reasoning chain of other examples) to better formulate their condition dependences.

Third, the gain of providing the reason seems to be moderate. From table 2, "judgement" (Figure 20 "question comparison") should be the key factor while "reason" (Figure 20, “problem decomposition” and “condition comparison”) seems to be less important. From Table 4, the proposed complicated RCoT seems to be worse than Self-Consistency (Wang et al., 2023).

**Questions:**

I expect the authors to justify their choice of interactively prompting the large language models. Currently, I felt they only proved it kind of works but did not explain the reason. However, I felt the comparison are not fair and the results are not easy to reproduce.

---

> ### Author Response · Authors · 2023-11-19
> **Response to reviewer rMGU**
>
> We thank the reviewer for the detailed and helpful feedback! We are happy that you find our method novel and effective! We address your concerns in the following texts:
>
> **W1:  the prompting method is overused**
>
> Thanks for raising this important point. We would like to clarify that “problem” and “question” are different in our paper. Specifically, “problem” = “conditions” + “question”. For example, a problem “I have two apples, and he has three apples. How many apples do we have in total?” = conditions [“I have two apples”, “he has three apples”] + the question “How many apples do we have in total?”. Therefore, to determine whether the reconstructed problem is identical to the original one, we first decompose problems into “conditions” (multiple conditions per problem) and “questions” (one question per problem), and then conduct “condition comparison” and “question comparison”. In short, “problem comparison” = “condition comparison” + “question comparison”. The final feedback is formulated based on the results of “problem comparison”.
> We believe there are some misunderstandings between caused by Fig 20. We apologize for the confusion and make a clearer Figure 20.  To rectify solutions, we do take into account “question comparison” results. However, in Fig 20, the “question comparison” results tell us that the question is correctly understood by models. Therefore, the final feedback in Fig 20 does not contain the result “question comparison” as the model did not make mistakes here. The final feedback will include the results of “question comparison” if mistakes are found in this process.
> We apologize for the confusion and add clarifications accordingly in the paper.
>
> **W2: the flaw of the "dialog system"**
>
> Our revision step only uses the history of original wrong answer, which is the same as Self-Refine[1]. The core of RCoT is to let LLMs check the answer and rectify it by themselves, so it is necessary to provide the previous answer. It’s worth noting that reconstruction, decomposition, comparison and revision are independent. Each stage only utilizes previous results but not conditions on previous prompts. We argue that “stateless” or not should not be the “flaw” of the method or experiment design, since all methods (RCoT, CoT, Self-Consistency, Self-Refine) only utilize LLM itself without external intervention. RCoT and Self-Refine enables LLM to act as humans that can check and revise answers by themselves to achieve the correct answers.
>
> [1] Madaan, Aman, et al. "Self-refine: Iterative refinement with self-feedback." arXiv preprint arXiv:2303.17651 (2023).
>
> **W3: the gain of providing the reason seems to be moderate**
>
> We hope our clarification in W1 could help address the understanding of Table 2. The “w/o reason” means we do not tell the LLM what exact mistakes it made, but only tell the LLM that it made mistakes. That is to say, as long as one error is found during “problem comparison”, we will tell the model it makes mistakes and needs to rectify its answers. Otherwise, we tell the model it does not make mistakes and the current answer is the final answer. The “w/o judgment + reason” has nothing to do with RCoT; it just tells models to double-check answers every time. Therefore, “w/o judgment + reason” denotes the necessity of giving coarse feedback (i.e., Wrong or Correct), whereas “w/o reason” further denotes that fine-grained feedback formulated from the results of “problem decomposition” can enhance the LLM’s capability of rectifying responses.
>
> We would also like to point out that RCoT performs better than Self-Consistency in 6 datasets and only underperforms Self-Consistency in one dataset in Table 4.

---

> > ### Author Response · Authors · 2023-11-22
> > **Official Comment by Authors**
> >
> > Dear reviewer rMGU,
> >
> > We appreciate your effort in reviewing our paper! Please let us know if we address your questions, and we are willing to have further discussions if needed:)

---

### Official Review · Reviewer_QEoj · 2023-10-30

**Soundness:** 3 good
**Presentation:** 3 good
**Contribution:** 3 good
**Rating:** 6
**Confidence:** 3

**Summary:**

This paper aims to tackle challenges like condition overlooking, question misinterpretation, and condition hallucination that LLMs meet in arithmetic reasoning benchmarks. On top of chain-of-thought (CoT) prompting, this paper proposes RCoT, which asks the LLM to rewrite the problem, compare it to the original one, and identify fine-grained differences in conditions and questions, thus finding mistakes and revising the original answer. Experiments show consistent improvements across benchmarks and LLMs, verifying effectiveness of the proposed method.

**Strengths:**

*Originality*: The core idea of this paper is novel and original.

*Quality*: The method is well-motivated and extensively evaluated.

*Clarity*: The delivery is very clear and easy to understand, I did not find issues in understanding.

*Reproducibility*: Code is provided to encourage reproducibility.

*Significance*: This work touches a major issue in LLMs, which is of much research significance.

**Weaknesses:**

- One drawback of this work could be its complexity, as illustrated in the diagram and verified by token counts. I understand it is comparable to some previous works, but optimizing its complexity is still an important aspect.
- (minor) The comparisons in tab. 1 is not very clear, eg, the results marked in green are not straightforward to understand except by reading the captions.
- (minor) Venues are missing from multiple references.

**Questions:**

Suggestion: it might be better to call "reconstruction" as "rewriting" or "paraphrasing".
Disclaimer: since I am not very familiar with related literature, my current rating is relatively conservative, and I'll reconsider it after reading opinions from other reviewers.

---

> ### Author Response · Authors · 2023-11-19
> **Response to reviewer QEoj**
>
> We thank Reviewer QEoj for the constructive comments. We are glad you find the RCOT method novel and effective. We will address your question in the following paragraphs.
>
> **W1: One drawback of this work could be its complexity**
>
> We understand the concern about the complexity of RCoT. However, as our experiments show, only fine-grained comparison could lead to fine-grained revision. See more detail in our general response W2.
>
> **W2 and W3:  The comparisons in tab. 1 is not very clear; Venues are missing from multiple references**
>
> We apologize for not clarifying some results and venues in the paper which led to difficulty in understanding. Specifically, we have already clarified the comparisons in Tab 1 and revised missed venues to help readers better understand RCoT.
>
> **Q1: it might be better to call "reconstruction" as "rewriting" or "paraphrasing"**
>
> Thank you for the helpful suggestion! Our idea is to get the original problem by reversing the answer, and find the factual inconsistency between the reversed problem and the original problem. Therefore, we think that "rewriting" or "paraphrasing cannot express our reversing ideas very well. We apologize for the confusion and re-clarify the concept in our paper.

---

> > ### Author Response · Authors · 2023-11-22
> > **Official Comment by Authors**
> >
> > Dear reviewer QEoj,
> >
> > We appreciate your effort in reviewing our paper! Please let us know if we address your questions, and we are willing to have further discussions if needed:)

---

> > > ### Comment · Reviewer_QEoj · 2023-11-22
> > >
> > > Thanks for the authors' response. Currently I have no more unresolved major concerns, and prefer to keep my positive recommendation.

---

### Author Response · Authors · 2023-11-19
**General Response**

We appreciate all reviewers for their efforts! We are glad that reviewers find our methods useful and novel. We address some common questions raised by reviewers below:

**GR1: Minor suggestions**

We highlighted our revisions with pink color in the paper. Specifically, we clarify the comparison in Table 1, revise missed venues, add additional information in Figure 20 to understand it easily, change the order of citing and unify Figures 1-3.

**GR2: The complex and incremental of RCoT**

Current complexities mainly come from the multi-step comparisons. The necessity of multi-step comparisons is due to the poor ability of LLMs on single-step comparison design. For example,  if we do not conduct condition comparisons one by one but ask LLMs to compare all conditions at once, the LLM will find it hard to reach the correct results. We believe LLMs with stronger entailment classification abilities can simplify the comparison design, thus boosting the inference.
As the difficulty of the problem increases, so does the complexity of the RCoT method. Intuitively, LLMs will perform poorly and make more mistakes for complex problems, and it is easier for RCoT to find mistakes (but it may be hard to find all mistakes). Therefore, the CoT may have lower performance, and RCoT may bring lower improvement for complex problems, which is also the case for many other methods (like self-critique and self-refine [1]), and more powerful LLMs can get more benefit from such methods (E.g., table 1 in our paper and table 1 in [1]).

The reviewer’s concern leads to an interesting direction on how we can generate feedback that is as good as humans since human feedback is more diverse, detailed, and accurate.

[1] Madaan, Aman, et al. "Self-refine: Iterative refinement with self-feedback." arXiv preprint arXiv:2303.17651 (2023).